# Pulmonary inflammation and viral replication define distinct clinical outcomes in fatal cases of COVID-19

Keyla S. G. de Sá[1¤], Luana A. Amaral[1], Tamara S. Rodrigues[1], Camila C. S. Caetano[1], Amanda Becerra[1], Sabrina S. Batah[2], Felipe T. Lopes[1], Isadora M. de Oliveira[1], Letícia S. Lopes[1], Leticia Almeida[1], Caroline M. Mota[1], Samuel Oliveira[1], Danilo T. Wada[3], Marcel Koenigkam-Santos[3], Ronaldo B. Martins[1], Roberta R. C. Rosales[1], Eurico Arruda[1], Alexandre T. Fabro[2], Dario S. Zamboni[1] *

1 Departamento de Biologia Celular e Molecular e Bioagentes Patogênicos, Faculdade de Medicina de Ribeirão Preto, Universidade de São Paulo, Ribeirão Preto, São Paulo, Brazil, 2 Departamento de Patologia e Medicina Legal, Faculdade de Medicina de Ribeirão Preto, Universidade de São Paulo, Ribeirão Preto, São Paulo, Brazil, 3 Departamento de Imagens Médicas, Hematologia e Oncologia, Faculdade de Medicina de Ribeirão Preto, Universidade de São Paulo, Ribeirão Preto, São Paulo, Brazil

¤ Current address: Department of Immunobiology, Yale School of Medicine, New Haven, Connecticut, United States of America
* dszamboni@fmrp.usp.br

**Data Availability Statement:** All relevant data are within the paper and its Supporting Information files.

## Abstract

COVID-19 has affected more than half a billion people worldwide, with more than 6.3 million deaths, but the pathophysiological mechanisms involved in lethal cases and the host determinants that determine the different clinical outcomes are still unclear. In this study, we assessed lung autopsies of 47 COVID-19 patients and examined the inflammatory profiles, viral loads, and inflammasome activation. Additionally, we correlated these factors with the patient's clinical and histopathological conditions. Robust inflammasome activation was detected in the lungs of lethal cases of SARS-CoV-2. Experiments conducted on transgenic mice expressing hACE2 and infected with SARS-CoV-2 showed that $Nlrp3^{-/-}$ mice were protected from disease development and lethality compared to $Nlrp3^{+/+}$ littermate mice, supporting the involvement of this inflammasome in disease exacerbation. An analysis of gene expression allowed for the classification of COVID-19 patients into two different clusters. Cluster 1 died with higher viral loads and exhibited a reduced inflammatory profile than Cluster 2. Illness time, mechanical ventilation time, pulmonary fibrosis, respiratory functions, histopathological status, thrombosis, viral loads, and inflammasome activation significantly differed between the two clusters. Our data demonstrated two distinct profiles in lethal cases of COVID-19, thus indicating that the balance of viral replication and inflammasome-mediated pulmonary inflammation led to different clinical outcomes. We provide important information to understand clinical variations in severe COVID-19, a process that is critical for decisions between immune-mediated or antiviral-mediated therapies for the treatment of critical cases of COVID-19.

**Funding:** Fundação de Amparo à Pesquisa do Estado de Sao Paulo (https://fapesp.br/), FAPESP grants 201/08216-2 (to DSZ), 2019/11342-6 (to DSZ) and 2020/04964-8 (to DSZ). Conselho Nacional de Desenvolvimento Científico e Tecnológico (https://www.gov.br/cnpq/pt-br), CNPq grant 03021/2020-9 (to DSZ). Coordenação de Aperfeiçoamento de Pessoal de Nível Superior (https://www.gov.br/capes/pt-br), CAPES grant 88887.50725/2020-00 (to DSZ). The funders had no role in study design, data collection and analysis, decision to publish, or preparation of the manuscript.

**Competing interests:** The authors have declared that no competing interests exist.

## Author summary

Although the SARS-CoV-2 pandemic has significantly affected the worldwide population and contributed to the death of millions worldwide, the processes that lead to patient death are still obscure. In this study, we assessed lung autopsies of 47 fatal cases of COVID-19 and found a significant activation of the NLRP3 inflammasome in the lungs of patients. To investigate whether this correlation between NLRP3 activation and disease exacerbation implies a cause-and-effect relationship, we performed infections in transgenic mice expressing the human ACE2 receptor. These mice were successfully infected with SARS-CoV-2, and we demonstrated that *Nlrp3*-deficient mice were protected from disease development and lethality compared to littermate animals sufficient for NLRP3, indicating a cause-consequence effect. Moreover, we found that patients who succumbed to COVID-19 can be divided into two main groups: Cluster 1, composed of patients who died with higher viral loads and reduced inflammation in the lungs, as opposed to patients from Cluster 2. We found that pulmonary thrombosis and disseminated intravascular coagulation were higher in patients from Cluster 1, contrasting with the significant development of fibrosis and hyperinflammation in Cluster 2 patients. Our study demonstrates the existence of two distinct profiles in lethal cases of COVID-19, a feature that is important to our understanding of lethality in this disease. Understanding these processes will be important to direct future therapeutic strategies for severe cases of COVID-19, either to target viral replication and thrombosis or to target the excessive inflammatory process.

## Introduction

After the COVID-19 pandemic, coronaviruses became frequent viral agents of acute respiratory distress syndrome (ARDS). Between 2019 and 2021, there were more than 255,000,000 confirmed cases and 5,127,000 deaths caused by the severe acute respiratory syndrome coronavirus 2 (SARS-CoV-2) worldwide [1]. Severe ARDS is characterized by major pulmonary involvement, respiratory distress, systemic thrombosis, and death [2–5]. The activation of the NLRP3 inflammasome has been described in response to SARS-CoV-2 infections [6–11], and this process contributes to excessive inflammation and poor clinical outcome. The recruitment of immune cells to the lungs of infected individuals culminates in the excessive release of cytokines that cause structural damage to the lungs [12,13]. Moreover, the major reported lung pathologies of severe COVID-19 include diffuse alveolar damage (DAD), acute fibrinoid organizing pneumonia (AFOP), and chronic interstitial pneumonia [2,3,5,14]. NLRP3 inflammasome activation also occurs in the lung parenchyma of these patients [11,15,16]. However, it is still unknown how inflammasome activation promotes the exacerbated inflammatory process during SARS-CoV-2 infection and how it impacts viral replication and the development of important clinical characteristics associated with death in COVID-19 patients. In the present study, postmortem lung tissue samples from 47 fatal COVID-19 patients were examined for inflammasome activation and gene expression; in addition, these factors were correlated with the clinical conditions of the patients to understand the molecular mechanisms underlying the pathological processes that lead to the death of these patients. An analysis of gene expression in pulmonary tissues allowed for the classification of COVID-19 patients into two different clusters: one cluster with patients who died with higher viral loads and reduced inflammatory profiles, which opposes the data from the other cluster. Our data establish that inflammasome-mediated pulmonary inflammation and viral replication define distinct clinical outcomes in fatal cases of COVID-19.

## Results

### Lethal cases of COVID-19 develop severe *acute respiratory distress syndrome* (ARDS) and higher inflammasome activation

We evaluated inflammasome activation in the lung parenchyma of 47 patients who died from SARS-CoV-2 infection from April to July 2020. These patients were infected with the ancestral strain of SARS-CoV-2 before the development of the COVID-19 variants of concern and before the use of vaccinations. As noninfected controls, we evaluated inflammasome activation in biopsies of the benign area of the lungs from patients who died due to lung adenocarcinoma (referred as uninfected controls). The data in **Table 1** show the demographic and clinical characteristics of the 47 patients. As reported in the early COVID-19 cases (that occurred in 2020), we found a high frequency of comorbidities, such as hypertension, obesity, diabetes, heart and lung disease. Furthermore, the patients exhibited altered values of CRP, D-dimers, LDH, creatinine, urea, AST, ALT, PT (INR), and blood glucose levels. The patients had an average illness time of 18 days, and the majority of the patients (76.59%) used mechanical ventilation (MV), with an average of 12.38 days of use of mechanical ventilation. The patients had considerably altered $PaO_2$ and $PaO_2/FiO_2$ characterizing a moderate to severe ARDS condition and consistent with the respiratory statuses. The histopathological analyses of COVID-19 patients demonstrated a high fibrotic phase of DAD, OP, and pneumonitis (**Table 1**). To analyze the histological features of patients' lungs, we measured the lung parenchyma area in the histological sections to assess loss of airspace. In support of the data described above, we observed that COVID-19 patients had a higher parenchyma area (loss of airspace), suggesting that SARS-CoV-2 triggers robust inflammatory infiltration to the lung in severe cases of the disease (**S1A–S1C Fig**). These data are in agreement with histopathological findings showing an intense inflammatory process in patients who died from SARS-CoV-2 [2,17]. To assess inflammasome activation in the lungs of COVID-19 patients, the lung parenchyma area and cell count were scanned in histological sections by using multiphoton microscopy. The parenchyma area was calculated by using ImageJ software, and inflammasome activation was microscopically scored by the presence of characteristic NLRP3 or ASC puncta/specks [18]. The parenchyma area was used to normalize all of the counts between the patients. Inflammasome activation was quantified by counting ASC and NLRP3 puncta in the lung tissue of these patients. We observed that patients with SARS-CoV-2 infection had robust inflammasome activation, as shown by the abundant presence of ASC and NLRP3 puncta in the lungs (**Fig 1A–1D**). Representative images of ASC and NLRP3 puncta in the patients' lungs are shown (**Fig 1B and 1D**). Importantly, we observed ASC colocalization in nearly all of the scored NLRP3 puncta, thus confirming that these puncta structures that were abundantly found in the patient's lungs are indeed the NLRP3/ASC inflammasome (**Fig 1E and 1F**). Collectively, these data indicate that COVID-19 patients have high inflammasome activation and that these patients evolved to a fibrotic phase of DAD, OP, and pneumonitis.

### Disease progression in fatal cases of COVID-19 occurs with decreasing viral load and increasing inflammasome activation

To assess whether inflammasome activation was associated with specific patient clinical conditions, we performed Pearson's correlations in COVID-19 patients. Even though all of these patients died, we observed a strong negative correlation between viral load and the time of the disease (symptoms onset to death) (**Fig 2A and 2B**). In addition, we observed a positive correlation between the amount of NLRP3 and ASC puncta with the time of disease (**Fig 2C and 2D**), thus suggesting that whereas the viral load is reduced, inflammasome activation increases

**Table 1. COVID-19 patient characteristics.**

| Demographic Characteristics | Mean (±SD) or N (%) | |
|---|---|---|
| N | 47 | |
| Sex | | |
| Female | 23 | (48.93%) |
| Age (years) | 67.97 | (15.05±) |
| Illness Time (days) | 18.08 | (11.04±) |
| ICU | 37 | (78.72%) |
| **Comorbidities** | | |
| Hypertension | 26 | (55.32%) |
| BMI | 31.10 | (±8.82) |
| Diabetes | 18 | (38.29%) |
| History of smoking | 12 | (25.53%) |
| Heart disease | 12 | (25.53%) |
| Lung disease | 12 | (25.53%) |
| Kidney disease | 7 | (14.89%) |
| History of stroke | 7 | (14.89%) |
| Autoimmune diseases | 2 | (4.25%) |
| **Laboratorial findings** | | |
| CRP (mg/dL) | 11.40 | (±8.49) |
| D-Dimers (µg/mL) | 4.91 | (±4.46) |
| LDH (mmol/L) | 5.01 | (±5.24) |
| Creatinine (mg/dL) | 2.22 | (±1.4) |
| Urea (mg/dL) | 119.40 | (±58.47) |
| AST (IU/L) | 104.42 | (±90.74) |
| ALT (IU/L) | 72.12 | (±55.14) |
| AST/ALT | 1.8 | (±1.60) |
| PT (INR) | 1.81 | (±2.13) |
| Albumin (g/dL) | 3.09 | (±0.58) |
| Blood Glucose (mg/dL) | 194.72 | (±94.60) |
| **Respiratory status** | | |
| Temperature (°C) | 37.39 | (±1.73) |
| Mechanical ventilation | 36 | (76.59%) |
| Nasal-cannula oxygen | 11 | (23.40%) |
| Intubated time (days) | 12.38 | (±7.14) |
| $P_aO_2$ (mmhg) | 74.01 | (±24.63) |
| Venous saturation ($S_vO_2$) | 67.58 | (±25.34) |
| $P_aO_2/FiO_2$[#] | 159.69 | (±90.84) |
| A-a $O_2$ Gradient Value[#] | 283.22 | (±208.31) |
| Respiratory Rate (mov/min) | 26.55 | (±6.80) |
| **Histopathological findings** | | |
| Fibrosis (% of area) | 30.00 | (±12.68) |
| Organizing Pneumonia (% of area) | 10.74 | (±14.89) |
| Acute Fibrinous and Organizing Pneumonia (% of area) | 6.91 | (±12.49) |
| Diffuse Alveolar Damage (% of area) | 17.87 | (±21.05) |
| Pneumonitis (% of area) | 22.57 | (±15.34) |

(*Continued*)

**Table 1.** (Continued)

| Demographic Characteristics | Mean (±SD) or N (%) | |
| --- | --- | --- |
| Pulmonary infarction (% of area) | 3.404 | (±9.618) |

\* Fisher's exact test

\*\* t-Test

\# data after 18 days of hospitalization

during hospitalization in lethal cases of COVID-19. We also observed significant negative correlations between NLRP3 puncta and $PaO_2/FiO_2$ (**Fig 2E**), suggesting that inflammasome activation is related to an overall worsening pulmonary function. We detected no statistically significant correlations between NLRP3 or ASC puncta and viral load in the tissues (**S2 Fig**).

## Pulmonary viral load and inflammatory gene expression define two patient clusters in lethal cases of COVID-19

The imbalance of inflammatory and anti-inflammatory processes leads to an excessive release of cytokines into the systemic circulation with potentially deleterious consequences, including systemic inflammatory response syndrome (SIRS), circulatory shock, multiorgan dysfunction syndrome (MODS), and death [19]. Several studies have reported the association of SARS-CoV-2 with hyperinflammatory syndrome related to disease severity [20–22], and inflammasome activation may be associated with Cytokine Release Syndrome (CRS) [23–25]. Although gene expression does not imply inflammasome activation, we assessed expression of inflammatory genes in the lungs of COVID-19 patients and found that several inflammatory cytokine genes positively correlate with genes of proteins involved in inflammasome activation (**S3A Fig**). We did not observe a significant difference in the expression of these genes when we compared COVID-19 patients with uninfected controls (**S3B–S3U Fig**), possibly due to the large dispersion of data observed in COVID-19 patients. Due to this dispersion and high variation in gene expression detected in COVID-19 patients, we performed an unsupervised heatmap constructed with data from relative gene expression and revealed the formation of two clusters in COVID-19 patients (**Fig 3**). Cluster 1 was characterized by a higher viral load and lower expression of inflammasome and inflammatory genes. In contrast to Cluster 1, Cluster 2 was comprised of patients who died with a lower viral load and increased expression of inflammasome and inflammatory genes (**Fig 3A**). To gain insights into key differences between these two clusters, we analyzed $PaO_2/FiO_2$ and A-a $O_2$ gradient kinetics in these patients. We observed that patients in Cluster 2, who had overall increased inflammation, had worsening pulmonary function compared to those in Cluster 1 (**Fig 3B and 3C**). Basic demographic information stratified by cluster is provided in **Table 2**. Strikingly, patients who belonged to Cluster 2 had a long illness time (**Fig 3D**), worse pulmonary function (as indicated by the $PaO_2/FiO_2$ and A-a $O_2$ gradients) (**Fig 3E and 3F**), greater inflammasome activation, as measured by NLRP3 puncta formation (**Fig 3G**), and increased area of pulmonary parenchyma, indicating loss of airways (**Fig 3H**). Images of the lung parenchyma of three representative patients from each cluster are shown (**Fig 3I and 3J**). Importantly, we found that patients belonging to Cluster 2 had an overall lower viral load than patients from Cluster 1; this was quantified both by RT-PCR for N2 and E gene expression (**Fig 4A and 4B**), and by expression of Spike protein (**Fig 4C–4E**). Next, we assessed radiological analyses of 15 patients from Cluster 1 and 31 patients from Cluster 2 by comparing the first and last chest X-ray images (CXR) and observed overall worsening pulmonary conditions in patients from Cluster 2 (**S4A Fig**). Representative images of CXR of two patients from Cluster 1, with Mild/Moderate pneumonia

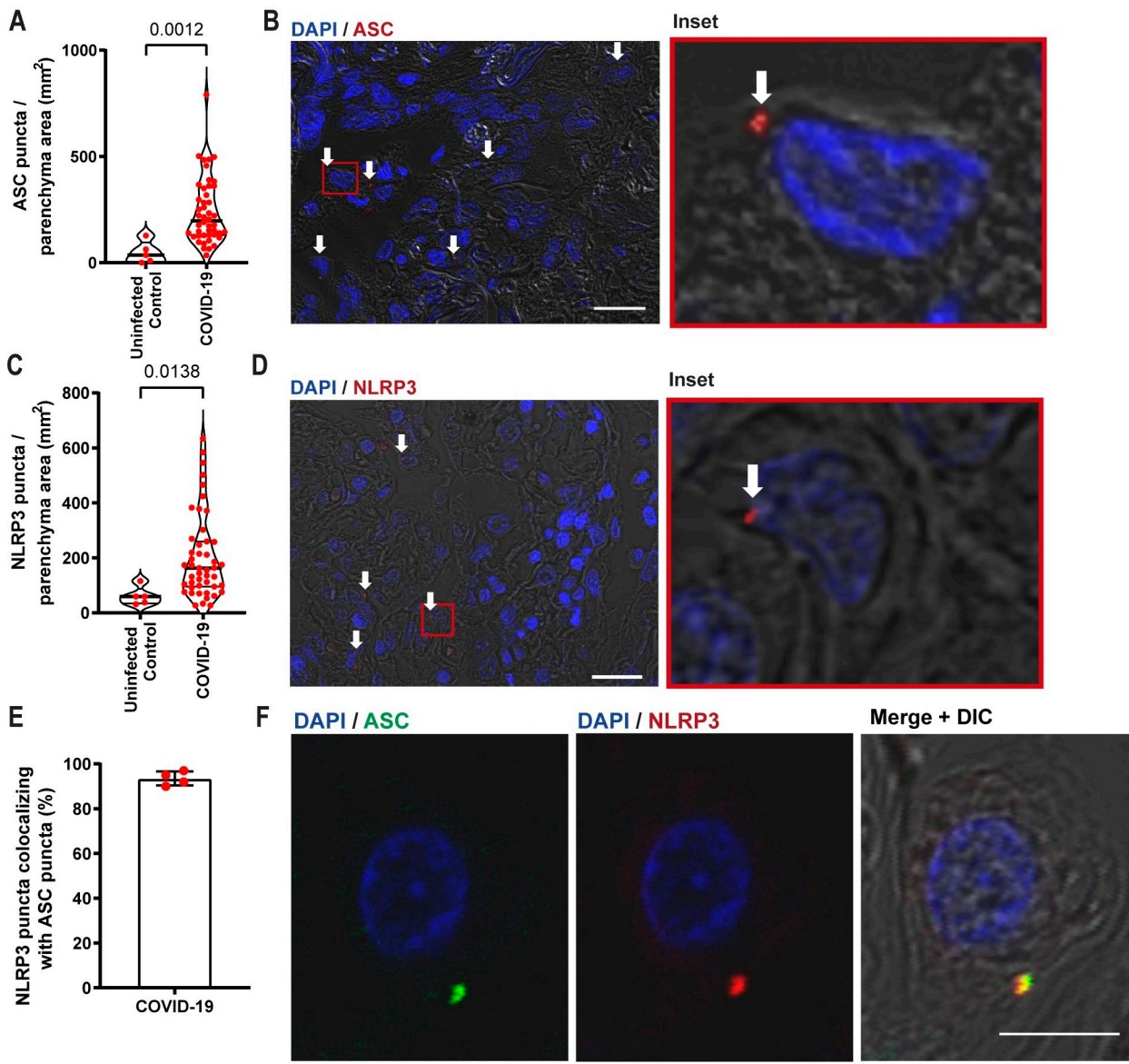

**Fig 1. NLRP3 Inflammasome activation in lung autopsy of COVID-19 patients.** Multiphoton microscopy analysis of lung autopsies of 47 COVID-19 patients and 5 uninfected controls. Tissues were stained with anti-ASC (**A, B**) or anti-NLRP3 (**C, D**) for quantification of cells with inflammasome puncta in lung autopsies (in red, indicated by white arrows). DAPI stains cell nuclei (blue). Insets indicate a higher magnification of the indicated region (red rectangle). Scale bars 20 μm. Each dot in the figure represents the value obtained from each individual. P-values are shown in the figures comparing the indicated groups, as determined by Mann–Whitney test. Data are represented as violin plots with median and quartiles. (**E**) Percentage of NLRP3 puncta colocalizing with ASC puncta in the lungs of five COVID-19 patients. (**F**) Representative images showing ASC (green) and NLRP3 (red) colocalization. Scale bar 10 μm. The images were acquired by multiphoton microscope using a 63x oil immersion objective and analyzed using ImageJ Software.

(stable conditions) (**S4B Fig**) and two patients from Cluster 2 with severe pneumonia (worsening) are shown (**S4C Fig**). **Table 3** summarizes the general imaging evaluation findings for the patients in Cluster 1 and Cluster 2. Taken together, our data suggest the existence of two distinct groups of patients who succumbed to COVID-19, one group with lower viral loads, higher inflammation, and worse pulmonary conditions than the other group.

We also tested whether the mechanical ventilation used in COVID-19 patients would interfere with inflammasome activation. This hypothesis is supported by data indicating that

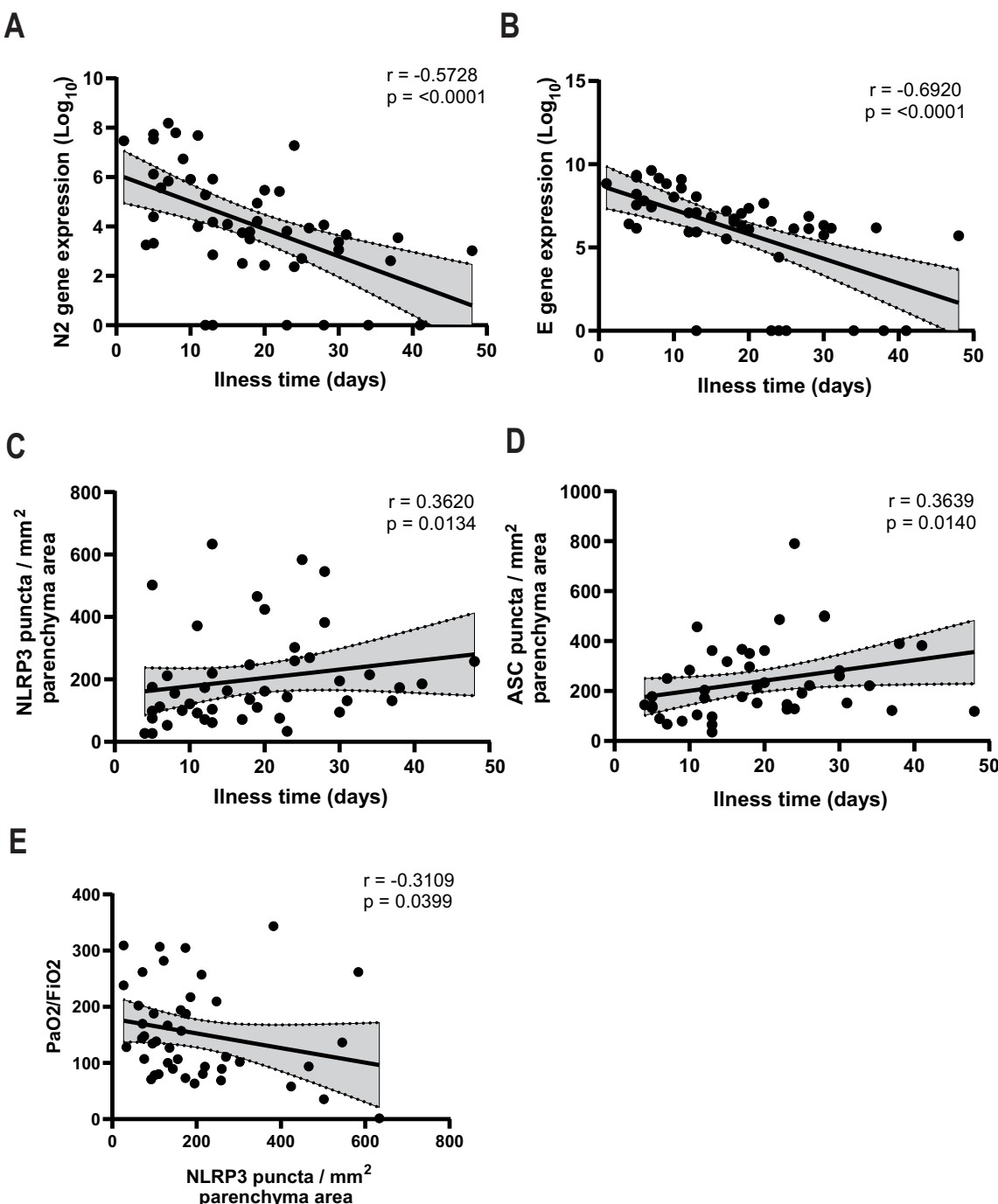

**Fig 2. NLRP3 inflammasome activation positively correlates with disease time, and viral load inversely correlates with disease time.** Spearman correlation of pulmonary viral load, inflammasome activation and illness time in 47 fatal COVID-19 patients. (**A**) Correlation of viral N2 with illness time; (**B**) Correlation of viral E with illness time; (**C**) Correlation of NLRP3 puncta per parenchyma area with illness time; (**D**) Correlation of ASC puncta per parenchyma area with illness time; (**E**) Correlation of NLRP3 puncta per parenchyma area with $PaO_2/FiO_2$. r and P-values are indicated in the figures.

mechanical ventilation-induced hyperoxia can induce potassium efflux through the $P_2X_7$ receptor, thus leading to inflammasome activation and the secretion of proinflammatory cytokines [24, 26–30]. However, in our study, when we separated the COVID-19 patients into two

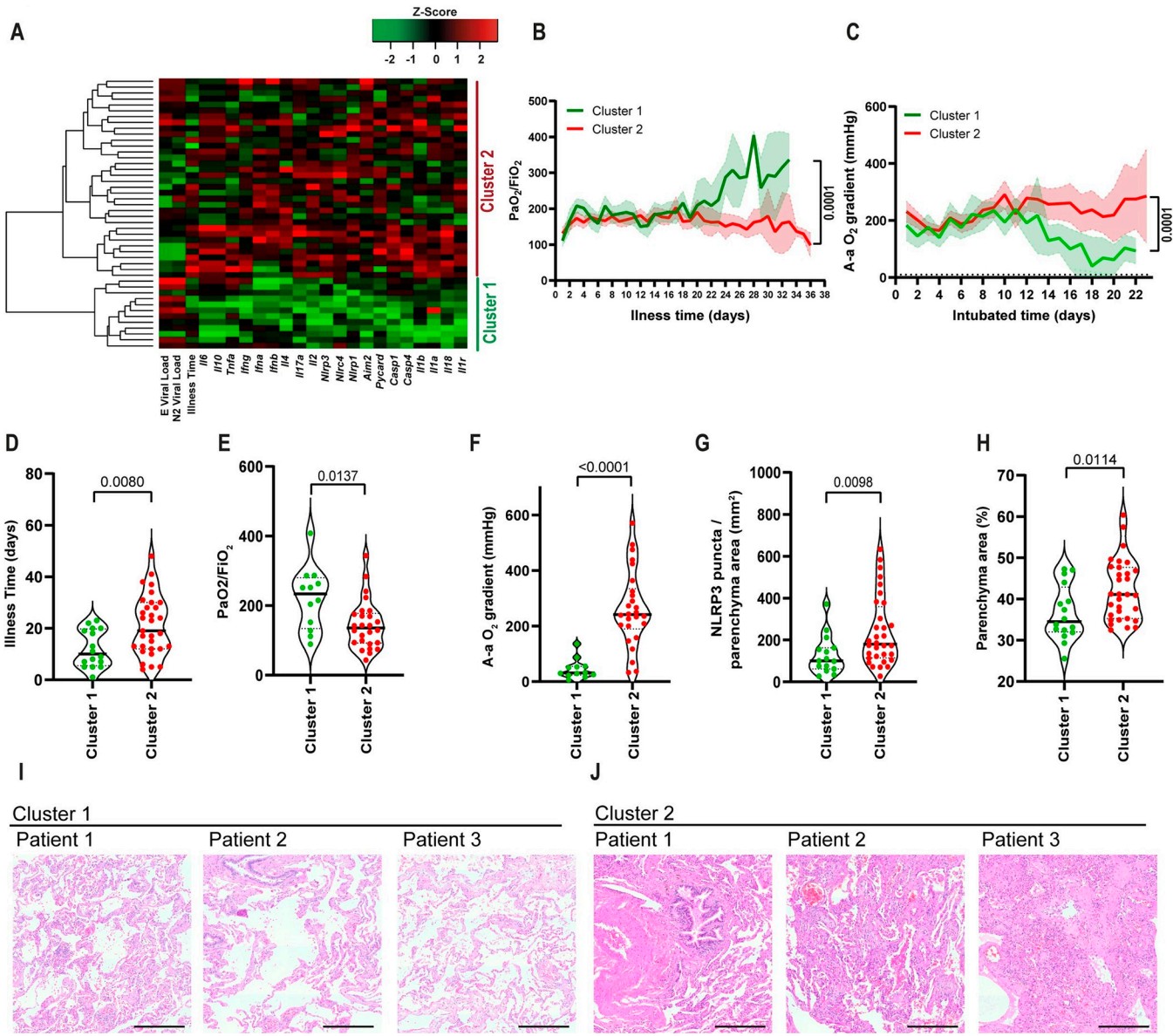

**Fig 3. Pulmonary viral load and inflammatory gene expression define two patient clusters in lethal cases of COVID-19.** (**A**) Heatmap of the mRNA expression of inflammasomes, inflammatory molecules/cytokines and viral N2 and E in lung autopsies of 47 COVID-19 patients. $PaO_2/FiO_2$ (**B**) and A-a $O_2$ gradient (**C**) during disease development of patients from Cluster 1 (N = 16, green) and Cluster 2 (N = 31, red). *, $P < 0.05$ comparing the indicated groups, as determined by Area Under Curve test. (**D-H**) Analysis of Cluster 1 and Cluster 2 for illness time (**D**), $PaO_2/FiO_2$ (**E**), A-a $O_2$ gradient (**F**), fibrosis (**G**), and NLRP3 puncta per lung parenchyma area (**H**). Each dot in the figure represents the value obtained from each individual. P-values are described in the figures comparing the indicated groups, as determined by Mann–Whitney test. Data are represented as violin plots with median and quartiles. (**I-J**) Representative H&E images of lung parenchyma of 3 patients of cluster 1 (**I**) and 3 patients of cluster 2 (**J**). Scale bars 200 μm.

groups according to the use or non-use of mechanical ventilation, we detected no differences in histopathological analyses and inflammasome activation (**Table 4**). Moreover, we confirmed that patients who underwent mechanical ventilation had a longer illness time, higher hypertension, less CRP and albumin, higher amounts of urea, and worsened pulmonary function (according to the PaO2/FiO2 and A-a O2 gradients). Thus, our data do not support the hypothesis that mechanical ventilation is directly associated with inflammasome activation.

**Table 2. COVID-19 patient characteristics.**

| Demographic Characteristics | Cluster 1 Mean (±SD) or N (%) | | Cluster 2 Mean (±SD) or N (%) | | *P* value |
|---|---|---|---|---|---|
| N | 16 | | 31 | | - |
| Sex | | | | | |
| Female | 6 | (37.50%) | 17 | (54.83%) | 0.358* |
| Male | 10 | (62.50%) | 14 | (45.16%) | |
| Age (years) | 70.81 | (±10.34) | 66.5 | (±17.46) | 0.369** |
| Illness Time (days) | 12.06 | (±6.97) | 21.19 | (±9.64) | 0.001** |
| ICU | 11 | (68.75%) | 25 | (80.64%) | 0.471* |
| **Comorbidities** | | | | | |
| Hypertension | 10 | (62.50%) | 16 | (51.61%) | 0.546* |
| BMI | 28.39 | (±6.92) | 33.21 | (±8.52) | 0.057** |
| Diabetes | 7 | (43.75%) | 11 | (35.48%) | 0.999* |
| History of smoking | 4 | (25.00%) | 8 | (25.80%) | 0.999* |
| Heart disease | 4 | (25.00%) | 8 | (25.80%) | 0.999* |
| Lung disease | 3 | (18.75%) | 9 | (29.03%) | 0.505* |
| Kidney disease | 1 | (6.25%) | 6 | (19.35%) | 0.395* |
| History of stroke | 3 | (18.75%) | 4 | (12.90%) | 0.675* |
| Autoimmune diseases | 1 | (6.25%) | 1 | (3.22%) | 0.999* |
| **Laboratorial findings** | | | | | |
| CRP (mg/dL) | 11.54 | (±1.64) | 11.32 | (±9.15) | 0.924** |
| D-Dimers (µg/mL) | 6.07 | (±5.45) | 4.38 | (±3.91) | 0.227** |
| LDH (mmol/L) | 6.17 | (±6.42) | 4.41 | (±4.52) | 0.280** |
| Creatinine (mg/dL) | 2.41 | (±1.64) | 2.12 | (±1.21) | 0.494** |
| Urea (mg/dL) | 108.36 | (±53.92) | 125.09 | (±60.75) | 0.358** |
| AST (IU/L) | 110.65 | (±112.67) | 101.08 | (±78.69) | 0.735** |
| ALT (IU/L) | 72.40 | (±59.92) | 71.98 | (±53.88) | 0.980** |
| AST/ALT | 2.1 | (±2.47) | 1.6 | (±0.92) | 0.319** |
| PT (INR) | 1.56 | (±0.98) | 1.95 | (±2.55) | 0.560** |
| Albumin (g/dL) | 3.24 | (±0.65) | 3.01 | (±0.54) | 0.251** |
| Blood Glucose (mg/dL) | 191.00 | (±100.25) | 196.65 | (±96.32) | 0.851** |
| Platelets ($10^3$/µl) | 173.70 | (±97.83) | 273.10 | (±128.20) | 0.010** |
| **Respiratory status** | | | | | |
| Temperature (˚C) | 37.74 | (±1.82) | 37.20 | (±1.62) | 0.304** |
| Mechanical ventilation | 11 | (68.75%) | 25 | (80.64%) | 0.471* |
| Nasal-cannula oxygen | 5 | (31.25%) | 6 | (19.35%) | 0.471* |
| Intubated time (days) | 8 | (±4.72) | 14.32 | (±6.94) | 0.009** |
| $P_aO_2$ (mmhg) | 75.54 | (±8.06) | 73.27 | (±1.69) | 0.135** |
| Venous saturation ($S_vO_2$) | 61.33 | (±27.83) | 70.59 | (±19.27) | 0.187** |
| $P_aO_2/FiO_2$[#] | 216.16 | (±32.94) | 164.15 | (±21.09) | <0.001** |
| A-a $O_2$ Gradient Value[#] | 40.20 | (±101.48) | 237.58 | (±68.05) | 0.006** |
| Respiratory Rate (mov/min) | 26.76 | (±7.86) | 26.45 | (±5.75) | 0.878** |
| PEEP | 9.63 | (±5.08) | 9.62 | (±4.86) | 0.9611** |
| **Histopathological findings, vascular status and Inflammasome activation** | | | | | |
| Fibrosis (% of area) | 24.38 | (±11.59) | 32.90 | (±14.05) | 0.042** |
| Organizing Pneumonia (% of area) | 23.30 | (±18.16) | 24.50 | (±15.71) | 0.815** |
| Acute Fibrinous and Organizing Pneumonia (% of area) | 18.33 | (±12.24) | 23.89 | (±15.76) | 0.225** |
| Diffuse Alveolar Damage (% of area) | 34.00 | (±24.03) | 33.33 | (±20.23) | 0.920** |

*(Continued)*

**Table 2.** (Continued)

| Demographic Characteristics | Cluster 1 Mean (±SD) or N (%) | | Cluster 2 Mean (±SD) or N (%) | | *P* value |
|---|---|---|---|---|---|
| Pneumonitis (% of area) | 20.67 | (±18.52) | 23.86 | (±15.88) | 0.540** |
| Pulmonary infarction (% of area) | 26.67 | (±14.14) | 20.00 | (±14.14) | 0.132** |
| Disseminated Intravascular Coagulation | 8 | (50.00%) | 5 | (16.13%) | 0.0198* |
| NLRP3 puncta per parenchyma area (mm$^2$) | 143.67 | (±89.00) | 238.40 | (±157.76) | 0.031** |
| Clearance Creatinine | 32.94 | (±22.95) | 47.50 | (±43.51) | 0.4380** |
| MDRD GFR | 41.58 | (±30.03) | 44.15 | (±36.90) | 0.9381** |
| Anticoagulation dose (mg) used close to death | 14.68 | (±10.07) | 9.85 | (±8.97) | 0.0125** |
| Glasgow scale | 7.18 | (±5.70) | 6.61 | (±5.23) | 0.7280** |
| SOFA score | 7.12 | (±2.30) | 7.25 | (±2.59) | 0.6828** |
| MELD score | 49.15 | (±21.88) | 50.91 | (±40.97) | 0.8113** |
| MODS | 10.57 | (±3.45) | 10.39 | (±3.16) | 0.9946** |

* Fisher's exact test

** t-Test

# data after 18 days of hospitalization

### Differential regulation of fibrinolysis-related genes and growth factors account for the development of fibrosis or disseminated intravascular coagulation in fatal cases of COVID-19

Histopathological analyses of the lungs from lethal cases of COVID-19 belonging to Clusters 1 and 2 allowed us to detect an increased development of fibrosis in patients from Cluster 2 (**Fig 5A** and **Table 2**). By contrast, according to the parameters recommended by the International Society of Thrombosis and Hemostasis (ISTH) [31] (additional details in Materials and Methods), patients belonging to Cluster 1 exhibited a markedly increased disseminated intravascular coagulation (**Fig 5B** and **Table 2**). In support of our analyses demonstrating increased disseminated intravascular coagulation, we found reduced platelet counts in patients from Cluster 1 (**Fig 5C** and **Table 2**), supporting that vascular dysfunctions impact clinical outcomes in patients belonging to Cluster 1.

To further investigate the biological pathways involved in the induction of fibrosis in Cluster 2 patients, we assessed the expression of fibrosis-related genes in the patient's lungs. We observed an increased expression of genes related to fibrinolysis, inflammatory cytokines, nitric oxide-induced, extracellular matrix deposition, and extracellular matrix remodeling in patients belonging to Cluster 2 (**Fig 5D**). Genes differentially expressed in samples from Cluster 2 compared to Cluster 1 include *Plg* (**Fig 5E**), *Plat* (**Fig 5F**), *Plau* (**Fig 5G**); *Ccr2* (**Fig 5H**), *Ccl3* (**Fig 5I**); *Cdkna1* (**Fig 5J**)*; Serpina1* (**Fig 5K**), *Timp2* (**Fig 5L**) and *Fgf1* (**Fig 5M**). To further illustrate the differences between fibrosis and thrombosis in patients from these different clusters, we performed Masson-Goldner staining to assess erythrocytes and clot formation (bright red) and collagen deposition (green) in Patients' lungs. By assessing three patients from each cluster, we found an increased clot formation in Cluster 1, suggesting an increased thrombotic process in these patients (**Fig 5N**). By contrast, samples from Cluster 2 show increased collagen deposition, suggesting an increased fibrotic process (**Fig 5O**). Together, these data support the hypothesis that fatal cases of COVID-19 progress via induction of disseminated intravascular coagulation, culminating in a thrombotic process or via a fibrotic process.

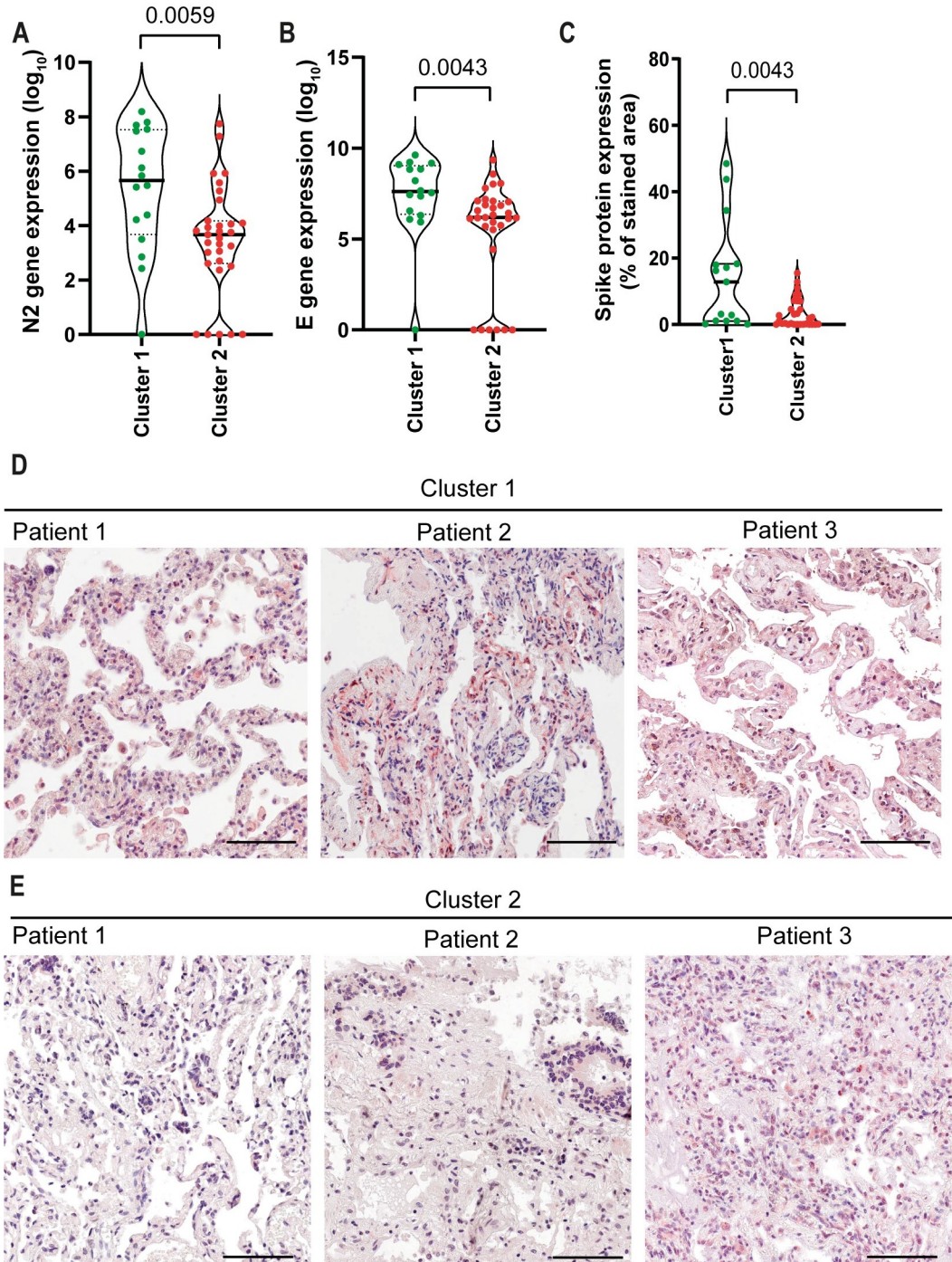

**Fig 4. COVID-19 patients from Cluster 1 contain higher viral loads in the lungs than patients from Cluster 2.**
Quantification of viral N2 (**A**) and E (**B**) in lung autopsies of 47 COVID-19 patients from Cluster 1 (N = 16) and Cluster 2 (N = 31). (**C**) Quantification of the percentage of area stained for viral Spike protein in lung autopsies. Each dot in the figure represents the value obtained from each individual. P-values are described in the figures comparing the indicated groups, as determined by Mann–Whitney test. Data are represented as violin plots with median and quartiles. (**D-E**) Representative images of lung tissues stained for Spike (red) and Hematoxylin (blue). Scale bars 100 μm.

**Table 3. Radiological Analyzes.**

| | Cluster 1 | | Cluster 2 | | P value |
|---|---|---|---|---|---|
| **Initial CXR pattern** | | | | | |
| No opacities | 1 | (6.250%) | 3 | (9.67%) | 0.9101 |
| Mild viral pneumonia | 2 | (12.50%) | 3 | (9.67%) | |
| Moderate viral pneumonia | 8 | (50.00%) | 12 | (38.70%) | |
| Severe viral pneumonia | 5 | (31.25%) | 11 | (35.48%) | |
| Impaired analysis | 0 | | 2 | (6.45%) | |
| **Last CXR pattern** | | | | | |
| No opacities | 1 | (6.25%) | 2 | (6.45%) | 0.9971 |
| Mild viral pneumonia | 3 | (18.75%) | 5 | (16.12%) | |
| Moderate viral pneumonia | 6 | (37.50%) | 12 | (38.70%) | |
| Severe viral pneumonia | 4 | (25.00%) | 8 | (25.80%) | |
| Impaired analysis | 2 | (12.50%) | 4 | (12.90%) | |
| **Imaging evolution until death** | | | | | |
| Improvement (fewer opacities) | 5 | (33.33%) | 10 | (32.25%) | 0.0364 |
| Stability | 7 | (46.66%) | 4 | (12.90%) | |
| Worsening (more opacities) | 2 | (13.33%) | 12 | (38.70%) | |
| Impaired analysis | 1 | (6.66%) | 5 | (16.12%) | |
| **Disease Extent (Thoracic CT)** | | | | | |
| 0 | 0 | | 0 | | 0.0363 |
| <25% | 4 | (66.66%) | 1 | (12.50%) | |
| 25–50% | 0 | | 3 | (37.50%) | |
| 50–75% | 1 | (16.66%) | 3 | (37.50%) | |
| >75% | 1 | (16.66%) | 1 | (12.50%) | |

## NLRP3 promotes disease pathology in ACE2-humanized mice infected with SARS-CoV-2

To investigate if NLRP3 inflammasome participates in the genesis of the inflammatory process induced by the SARS-CoV-2 or if it is only an additional marker of inflammation, we performed in vivo infections in a mouse model of COVID-19 [32]. We generated a mouse strain expressing human ACE2 that is sufficient or deficient in NLRP3. Mice were infected intranasally with $2x10^4$ PFU, and disease parameters were evaluated 3 or 5 days after infection (**Fig 6A**). Initially, we evaluated the pulmonary status of SARS-CoV-2-infected mice. Histological analyses performed by assessing H&E-stained sections of the lung tissues indicated a reduced parenchyma area as compared to $Nlrp3^{+/+}$ mice infected for 3 (**Fig 6B**) and 5 days post-infection (**Fig 6C**), corroborating our findings that patients with more inflammasome activation have increased parenchyma area, worse lung function, and severe pneumonia. Next, we investigated the effect of NLRP3 on disease development. Initially, we assessed the weight loss and found that $Nlrp3^{-/-}$ mice show a slight protection in weight loss compared to $Nlrp3^{+/+}$ littermate controls (**Fig 6D**). $Nlrp3^{-/-}$ mice show also a reduced clinical score during disease development (**Fig 6E**). To evaluate mice mortality during disease development, we monitored mice for up to 14 days after infection and found that SARS-CoV-2-infected $Nlrp3^{-/-}$ were significantly more resistant to death as compared to $Nlrp3^{+/+}$ controls (**Fig 6F**). Fig 6D–6F are a pool of two independent experiments with similar results, comprehending a total of n = 14 $Nlrp3^{+/+}$ and n = 15 $Nlrp3^{-/-}$. Collectively, these data indicate that NLRP3 affects the disease progression in mice, supporting the hypothesis (and clinical findings) indicating that NLRP3 is critical for the genesis of severe inflammation observed in some lethal cases of COVID-19.

**Table 4. Characteristics of COVID-19 patients who were submitted or not to mechanical ventilation (MV).**

| Demographics Characteristics | MV+ Mean (±SD) or n (%) | | MV- Mean (±SD) or n (%) | | p value |
|---|---|---|---|---|---|
| N | 36 | | 11 | | |
| Sex | | | | | |
| Female | 17 | (47.22%) | 6 | (54.54%) | 0.7400* |
| Male | 19 | (52.77%) | 5 | (45.45%) | |
| Age (years) | 65.61 | (±13.52) | 75.72 | (±17.61) | 0.0493** |
| Illness Time (days) | 20.94 | (±10.76) | 8.72 | (±5.40) | 0.0007** |
| ICU | 36 | (100.00%) | 0 | (0.00%) | >0.9999* |
| **Comorbidities** | | | | | |
| Hypertension | 23 | (63.88%) | 3 | (27.27%) | 0.0433* |
| BMI | 30.61 | (±8.52) | 34.56 | (±14.46) | 0.4324** |
| Diabetes | 16 | (44.44%) | 2 | (18.18%) | 0.1644* |
| History of smoking | 11 | (30.55%) | 3 | (27.77%) | >0.9999* |
| Heart disease | 11 | (30.55%) | 1 | (9.09%) | 0.2440* |
| Lung disease | 9 | (25.00%) | 2 | (18.18%) | >0.9999* |
| Kidney disease | 6 | (16.66%) | 1 | (9.09%) | >0.9999* |
| History of stroke | 4 | (11.11%) | 3 | (27.77%) | 0.3296* |
| Autoimmune diseases | 1 | (2.77%) | 1 | (9.09%) | 0.4172* |
| **Laboratorial findings** | | | | | |
| CRP (mg/dL) | 9.99 | (±7.83) | 16.02 | (±9.29) | 0.0377** |
| D-Dimers (µg/mL) | 4.99 | (±4.68) | 4.56 | (±3.62) | 0.7989** |
| LDH (mmol/L) | 5.78 | (±5.76) | 2.49 | (±1.13) | 0.0680** |
| Creatinine (mg/dL) | 1.58 | (±1.39) | 1.05 | (±0.50) | 0.2242** |
| Urea (mg/dL) | 138.18 | (±54.07) | 64.48 | (±34.12) | 0.0001** |
| AST (IU/L) | 109.61 | (±98.16) | 84.81 | (±54.14) | 0.4725** |
| ALT (IU/L) | 79.02 | (±58.67) | 45.38 | (±26.97) | 0.1254** |
| AST/ALT | 1.6 | (±1.74) | 2.2 | (±0.88) | 0.3542** |
| PT (INR) | 1.53 | (±0.91) | 2.82 | (±4.24) | 0.0900** |
| Albumin (g/dL) | 2.99 | (±0.54) | 3.78 | (±0.41) | 0.0034** |
| Blood Glucose (mg/dL) | 198.22 | (±93.77) | 181.11 | (±102.34) | 0.6340** |
| **Respiratory status** | | | | | |
| Temperature (˚C) | 37.51 | (±1.79) | 36.99 | (±1.53) | 0.3890** |
| $P_aO_2$ (mmhg) | 79.57 | (±20.19) | 53.99 | (±24.12) | 0.0014** |
| Venous saturation ($S_vO_2$) | 70.86 | (±14.36) | 50.71 | (±31.26) | 0.0092** |
| $P_aO_2/FiO_2$[#] | 132.63 | (±67.94) | 257.11 | (±114.86) | <0.0001** |
| A-a $O_2$ Gradient Value[#] | 351.09 | (±183.40) | 45.69 | (±26.55) | <0.0001** |
| Respiratory Rate (mov/min) | 27.61 | (±6.17) | 23.09 | (±7.89) | 0.0669** |
| **Histopathological findings** | | | | | |
| Fibrosis (% of area) | 30.28 | (±14.05) | 29.09 | (±12.21) | 0.8015** |
| Organizing Pneumonia (% of area) | 26.11 | (±13.56) | 20.00 | (±11.54) | 0.4520** |
| Acute Fibrinous and Organizing Pneumonia (% of area) | 19.50 | (±12.48) | 26.00 | (±15.16) | 0.3906** |
| Diffuse Alveolar Damage (% of area) | 28.82 | (±19.34) | 43.75 | (±16.85) | 0.0742** |
| Pneumonitis (% of area) | 23.52 | (±17.33) | 20.00 | (±8.16) | 0.5435** |
| Pulmonary infarction (% of area) | 25.00 | (±10.00) | 10.00 | (±0.00) | 0.2235** |
| **Inflammasome activation** | | | | | |
| NLRP3 puncta per parenchyma area ($mm^2$) | 204.3 | (±155.5) | 204.0 | (±156.6) | 0.9310** |

(*Continued*)

**Table 4.** (Continued)

| Demographics Characteristics | MV+ Mean (±SD) or n (%) | | MV- Mean (±SD) or n (%) | | p value |
|---|---|---|---|---|---|
| ASC puncta per parenchyma area (mm²) | 251.8 | (±192.5) | 267.3 | (±133.1) | 0.4196** |

\* Fisher's exact test

\*\* t-Test

\# data after 18 days of hospitalization

## Discussion

COVID-19 is significantly lethal in nonvaccinated individuals, and although inflammation and cytokine storm are associated with poor clinical outcomes, the mechanisms underlying dysregulated inflammatory processes are unknown. The revelation that exacerbated inflammasome activation contributes to COVID-19 pathology [8,9,11] advanced the understanding of disease pathology, but a significant proportion of the lethal cases progresses with lower inflammasome activation. Our analysis of 47 fatal COVID-19 cases allowed for the classification of the patients into two groups. Cluster 2 showed a remarkably high inflammasome activation and hyperexpression of inflammatory genes with increased pulmonary fibrosis and worsened respiratory functions. By contrast, patients belonging to Cluster 1 died faster, with higher viral loads, reduced inflammatory process, and increased disseminated intravascular coagulation. Our data reveal two distinct profiles in lethal cases of COVID-19, thus indicating that the balance of viral replication and inflammasome-mediated pulmonary inflammation led to different clinical outcomes.

Patients belonging to Cluster 2, died of poor respiratory functions and increased fibrosis induced by the excessive inflammatory process. Inflammatory cytokines released upon inflammasome activation have been previously linked to the development of pulmonary fibrosis [32–37]. In addition, cytokines such as IL-1β, IL-18, and IL-1α have been described as triggering the activation of fibroblasts and stimulating the synthesis and accumulation of type I collagen, TIMP, collagenase, and PGE2 [33,38], which may contribute to the triggering of the deleterious effects of inflammasomes in patients' lungs. Interestingly, patients from Cluster 2 died with low viral loads and higher inflammasome activation; in many patients from this cluster, viral loads were not detected, suggesting that SARS-CoV-2 per see may not be required for continuous inflammasome activation in the lungs of these patients. How NLRP3 inflammasome is activated in the tissues is still unknown, it is possible that Damage-Associated Molecular Patterns (DAMPs) participate in this process. Alternatively, it is possible that viral proteins produced before viral elimination or undetectable levels of SARS-CoV-2 present in the tissues are sufficient to activate the NLRP3 inflammasome. Future studies will be required to clarify these questions. Features observed in patients from cluster 1 are consistent with the previously reported damage in lung epithelial, endothelial, alveolar damage and thrombosis in fatal COVID-19 [39–41]. It would be interesting to evaluate the status of inflammasome activation in these patients, but one would predict low activation of the inflammasome and high viral loads, consistent with the features observed in cluster 1.

In this study, we identified two distinct profiles in lethal cases of COVID-19. Experiments performed with NLRP3-deficient mice support our assertion that NLRP3 inflammasome indeed participates in the genesis of the inflammatory process observed during COVID-19, as opposed to the hypothesis that NLRP3 is only an additional marker of inflammation. The role of NLRP3 in the regulation of fibrinolysis-related genes, which are elevated in cluster 2 patients is a matter for future investigation, but it is possible that activation of this inflammasome

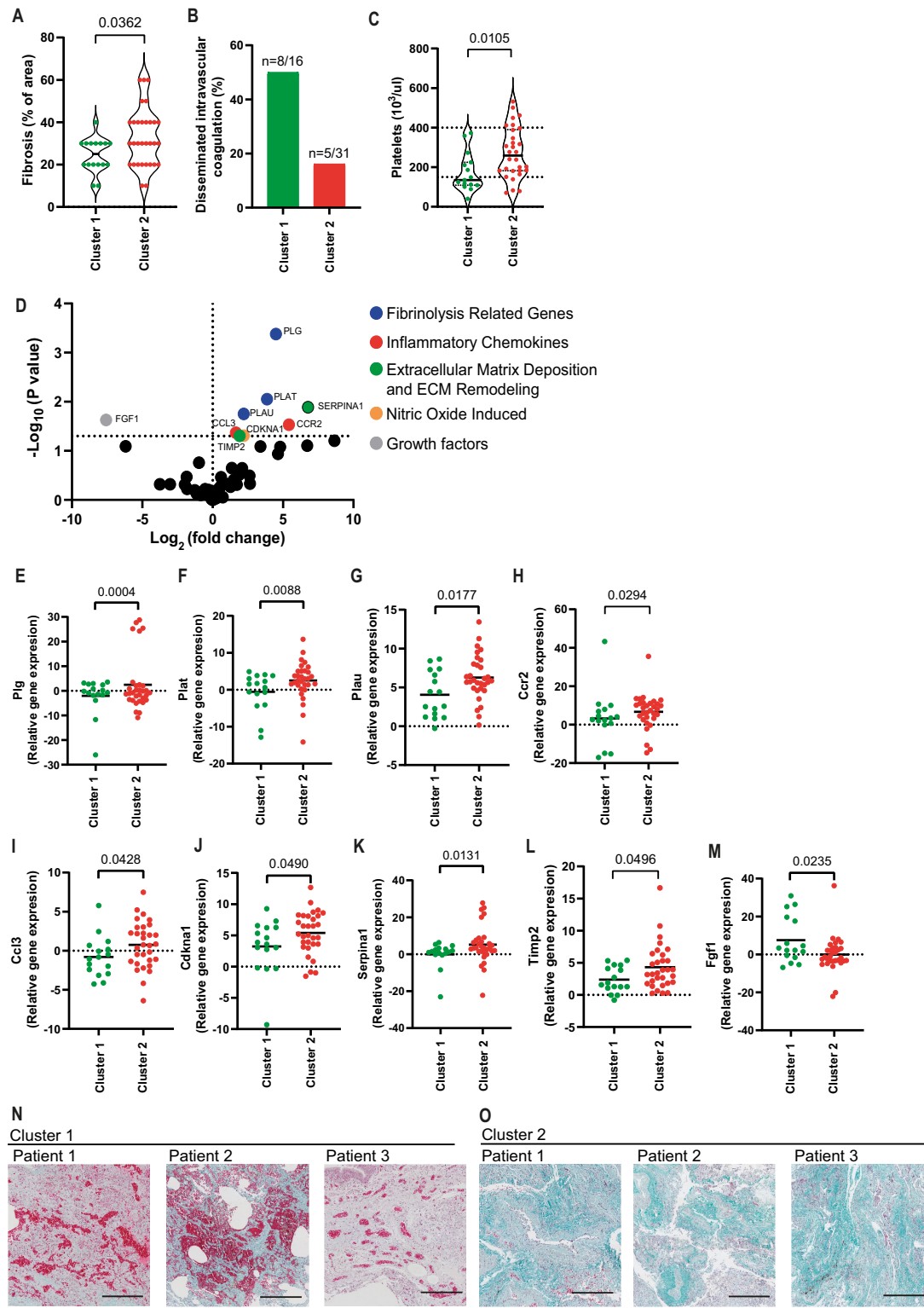

**Fig 5. Fibrosis and disseminated intravascular coagulation are differentially observed in the two patient clusters. (A-B)** Histopathological analysis of pulmonary samples from Cluster 1 and Cluster 2 for the presence of fibrosis (**A**) and Disseminated Intravascular Coagulation (**B**). (**C**) Platelet counts by laboratory analyses of blood samples from COVID-19 patients on the last day of hospitalization. (**D**) Fold change (Cluster2/Cluster1) expression of fibrosis-related genes in lung autopsy of COVID-19 patients. Each dot in this figure represents a gene (average gene expression of 47 patient samples). The genes with statistically significant

differences are indicated in color dots: *Plg* (**E**), *Plat* (**F**), *Plau* (**G**), *Ccr2* (**H**), *Ccl3* (**I**), *Cdkna1* (**J**), *Serpina1* (**K**), *Timp2* (**L**), and *Fgf1* (**M**). Each dot in the figures (**A, C, E-M**) represents the value obtained from each individual. P-values are described in the figures comparing the indicated groups, as determined by Mann–Whitney test. Data are represented as violin plots with median and quartiles. (**N-O**) Representative images of Masson-Goldner staining of lung parenchyma from three Cluster 1 patients (**N**) and three Cluster 2 patients (**O**). Shown are Nuclei (dark brown to black), Collagen (green/blue), Erythrocytes (bright red). Scale bars 200μm.

promotes the genesis of fibrosis observed in cluster 2 patients. Regardless of the role of NLRP3 inflammasome in the induction of fibrotic process, the revealed balance of viral-induced intra-vascular coagulation versus inflammasome-mediated pulmonary inflammation contributes to our understanding of disease pathophysiology and may contribute to decisions between immune-mediated or antiviral-mediated therapies for the treatment of critical cases of COVID-19.

## Methods

### Ethics statement

This study was approved by the Research Ethics Committee of Faculdade de Medicina de Ribeirão Preto, Universidade de São Paulo under protocol n° 4,089.567. Patients' consent was not required because this study is retrospective, using paraffin blocks containing lung tissue available at the institutional Service of Pathology, SERPAT.

### Post-Mortem samples

Minimally invasive autopsy was performed on 47 patients diagnosed with SARS-CoV-2 at the Hospital das Clínicas da Faculdade de Medicina de Ribeirão Preto da Universidade de São Paulo, Brazil (Ribeirão Preto, SP, Brazil) from April to July, 2020, by Serviço de Patologia (SERPAT) of the Hospital das Clínicas, Departamento de Patologia da Faculdade de Medicina de Ribeirao Preto, Universidade de São Paulo. Minimally invasive autopsies were done at the bedside through post-mort surgical lung biopsy by a matching 14-gauge cutting needle (Magnum Needles, Bard) and a biopsy gun (Magnum, Bard). Moreover, a 3 cm incision on the more affected side of the chest between the fourth and fifth ribs were also used to provide extra lung tissue. All tissue samples were embedded in paraffin and fixed in formalin (Formalin-Fixed Paraffin- Embedded, FFPE). Lung tissue from biopsy of Lung Adenocarcinoma patients (n = 5) were obtained from SERPAT.

### Immunofluorescence and histological evaluation

The slides were incubated with the primary antibodies, rabbit anti-human NLRP3 mAb (clone D2P5E; 1:300; Cell Signaling), rabbit anti-human ASC polyclonal antibody (1:200; Adipogen AL177), overnight at 4°C and with the secondary antibodies Goat anti-mouse Alexa fluor-647 (Invitrogen) or Goat anti-rabbit Alexa fluor-594 (Invitrogen). Images were acquired by the Axio Observer system combined with the LSM 780 confocal device microscope at 63x magnification (Carl Zeiss). Paraffin-embedded lung tissues sections (3 μm) were stained by standard hematoxylin and eosin (H&E). Morphological lung injury patterns were evaluated by specialized pulmonary pathologists (ATF) blinded to clinical history. They were classified as absent or present with its extent of lung injury area by 5% cut-offs in Fibrosis, Organizing Pneumonia (OP), Acute Fibrinous and Organizing Pneumonia (AFOP), Diffuse Alveolar Damage (DAD), Cellular Pneumonitis, Thrombus Formation. Evaluation of Disseminated Intravascular Coagulation (DIC) was performed according to the recommendation by The International Society on Thrombosis and Hemostasis (ISTH) [31]. This score consists of evaluating the following

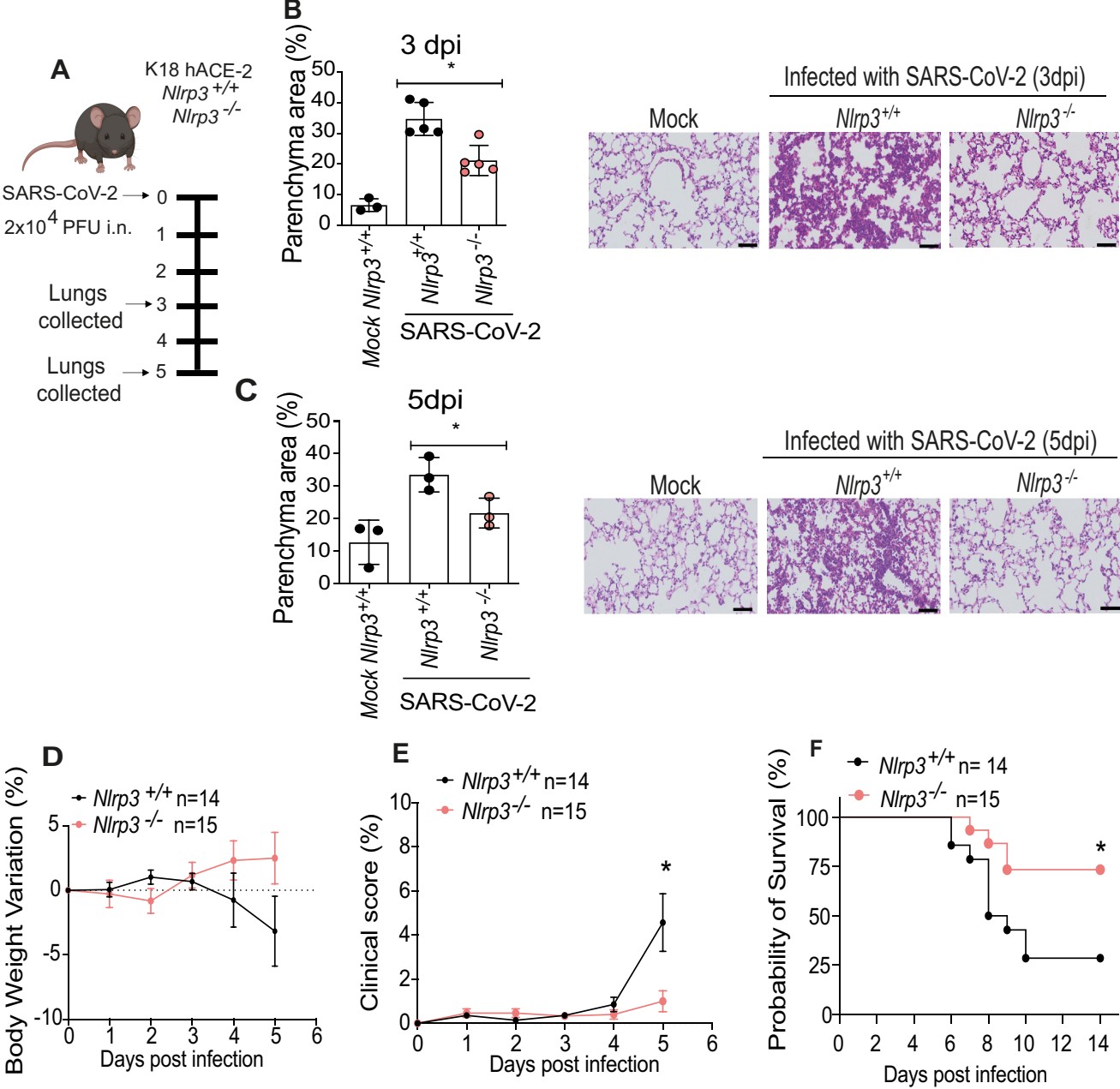

**Fig 6. NLRP3 promotes disease exacerbation in ACE2-humanized mice infected with SARS-CoV-2.** Transgenic K18-hACE2 mice deficient of sufficient for NLRP3 were infected intranasally with $2x10^4$ SARS-CoV-2 per mouse (A). The percentage of parenchyma area was estimated after hematoxylin and eosin (H&E) staining of lung sections of K18-hACE2 $Nlrp3^{+/+}$ and $Nlrp3^{-/-}$ infected for three (B) or five days after infection (C). Representative images of H&E-stained lungs of mice infected for three (B) and five (C) days are shown. Body weight variation (D) and clinical score (E) were measured daily for 5 days after infection. Survival curve of K18-hACE2 $Nlrp3^{+/+}$ (n = 14), and K18-hACE2 $Nlrp3^{-/-}$ (n = 15) infected mice and followed for 14 days (F). * $P < 0.05$, as determined by t test (B-E) or by Kaplan-Meier (F). Mock-treated mice (non-infected) or SARS-CoV-2 infected mice are indicated in the figure. Fig 6D–6F are a pool of two independent experiments with similar results, comprehending a total of n = 14 $Nlrp3^{+/+}$ and n = 15 $Nlrp3^{-/-}$ mice. Scale bar 50 μm. Fig 6A was created with BioRender.com.

clinical variables: platelet count ($>100$ = 0;$<100$ = 1;$<50$ = 2); elevated fibrin-related marker such as soluble fibrin monomers/fibrin degradation products (no increase = 0; moderate increase = 2; strong increase = 3); prolonged prothrombin time ($<$ 3 sec. = 0; $>$ 3 sec. but $<$ 6 sec. = 1; $>$ 6 sec. = 2); fibrinogen level ($>$ 1.0 gram/l = 0; $<$ 1.0 gram/l = 1). If the final sum of values is greater than or equal to 5, the patient is considered positive for DIC.

## RNA extraction and Real-Time Polymerase Chain Reaction for inflammatory genes

Total RNA from fresh lung tissue of SARS-CoV-2 patients and controls was obtained using Trizol reagent, and purification was performed according to the manufacturer's instructions. The RNA was quantified by spectrophotometry in a NanoDrop 2000c spectrophotometer. The concentration was adjusted to 1 μg/μL, and the RNA was stored at −70°C until reverse transcription. For inflammatory genes qPCR the total RNA was transcribed into complementary DNA (cDNA) using a High-Capacity cDNA Reverse Transcription kit (without an inhibitor) according to the protocol provided by the manufacturer (Thermo Fisher, Carlsbad, CA, USA). The reaction was prepared in a final volume of 20.0 μL containing 4.2 μL of $H_2O$, 2.0 μL of buffer, 2.0 μL of random primers, 0.8 μL of dNTP Mix (100 mM), 1.0 μL of reverse transcriptase (RT) enzyme and 1 μL of RNA (1 ug/μL). The solution was then placed into a thermocycler with the following program: 25°C for 10 min, 37°C for 120 min and 85°C for 5 min. The real-time PCR was performed in 96-well plates using Sybr Green reagents (Applied Biosystems, Waltham, MA, USA) and a Quant studio real-time PCR system (Applied Biosystems, Foster City, CA, USA). The real time-RT-PCR was carried out in a total volume of 20 μl on a 96-well MicroAmp Fast Optical plate (Applied Biosystems). Each well contained 10 μl SYBR Green qPCR Master Mix (Thermofisher), 1 μl of each primer (**S1 Table**), 2 μl cDNA (20 ng) and 7 μl RNase free water using the following protocol: initial denaturation at 95°C for 10 min, 40 cycles of denaturation at 95°C for 15 s followed by annealing/extension at 60°C for 60 s. Each PCR was followed by a dissociation curve analysis between 60–95°C. The Ct values were analyzed by the comparative Ct (ΔΔCt) method and normalized to the endogenous control GAPDH. Fold difference was calculated as $2^{-\Delta\Delta Ct}$

## Real-Time Polymerase Chain Reaction for Viral RNA

Detection and quantification of SARS-CoV-2 genes was performed with primer-probe sets for 2019-nCoV_N2 and gene E, according to US Centers for Disease Control and Prevention [42] and Charité group protocols [43]. The genes evaluated (N2, E, and RNase-P housekeeping gene) were tested by one-step real-time RT-PCR using total nucleic acids extracted with TRIzol (Invitrogen). All real-time PCR assays were done on a Quant studio real-time PCR system (Applied Biosystems, Foster City, CA, USA). A total of 70 ng of RNA was used for genome amplification, adding specific primers (20 μM), and probe (5 μM), and with TaqPath 1-Step quantitative RT-PCR Master Mix (Applied Biosystems), with the following parameters: 25°C for 2 min, 50°C for 15 min, and 95°C for 2 min, followed by 45 cycles of 94°C for 5 s and 60°C for 30 s. Primers used were the following: N2 forward: 5′-TTACAAACATTGGCCGCAAA-3′, N2 reverse: 5′-GCGCGACATTCCGAAGAA-3′; N2 probe: 5′-FAM-ACAATTTGCCCC-CAGCGCTTCAG-BHQ1-3′ [42]; E forward: 5′-ACAGGTACGTTAATAGTTAATAGCGT-3′, E reverse: 5′-ATATTGCAGCAGTACGCACACA-3′; E probe: 5′-AM-ACACTAGC-CATCCTTACTGCGCTTCG-BHQ-1-3′ [43]; RNase-P forward: 5′-AGATTTGGACCTGC-GAGCG-3′, RNase-P reverse: 5′-GAGCGGCTGTCTCCACAAGT-3′; and RNase-P probe: 5′-FAM-TTCTGACCTGAAGGCTCTGCGCG-BHQ-1-3′ [42]. A plasmid of N2 protein and E protein was used for a standard curve construction for viral load quantification.

## Chest computed tomography images acquisition and evaluation

Chest x-radiography (CXR) and computed tomography (CT) exams were performed as part of the routine clinical evaluation. Chest radiographies were performed in conventional equipment, mainly in the anteroposterior incidence. CT images were performed in multidetector scanners (Brilliance CT Big Bore 16—Philips, Holland, or Aquilion Prime 160—Toshiba, Japan), using similar protocols for the acquisition of high-resolution images of the lungs [44]. Patients were scanned in the supine position without the administration of intravenous contrast media. Typical acquisition parameters were: 120 kVp tube voltage, 100–140 ref mAs [45], 0.3–0.7 s gantry rotation time, reconstruction matrix size of 512×512, slice thickness, and increment of 1.0 mm, using standard (soft) and hard kernel filters. Imaging exams were independently evaluated by two thoracic radiologists (MKS and DTW), blinded to clinical data, laboratory, and pathology results as described [45]. Divergences were solved by consensus. All CXRs available were classified for the presence and grade of viral pneumonia [46]. Pulmonary disease evolution on imaging was evaluated considering all exams from the initial to the last image before death and classified as: radiological improvement, stability, worsening, or probable secondary complication. Tomographic images were evaluated similarly to CXR images.

## Mouse and in vivo infections

All mice experiments were conducted according to the institutional ethical committees for animal care (Comissão de Ética em Experimentação Animal da Faculdade de Medicina de Ribeirão Preto FMRP/USP), approved protocol number 133/2020. The mice used in this study were in C57BL/6 genetic background and included K18-hACE2 (B6.Cg-Tg(K18-ACE2) 2Prlmn/J; JAX strain # 034860) that were crossed with *Nlrp3* deficient strain (JAX strain # 017969) for generation of $hACE2^{+/-}Nlrp3^{+/-}$, which were crossed for the generation of the $hACE2^{+/-}Nlrp3^{+/+}$, $hACE2^{+/-}Nlrp3^{+/-}$, and $h-ACE2^{+/-}Nlrp3^{-/-}$. Female mice ranging from eight- to ten-weeks-old were infected in a BSL3 facility at the University of São Paulo, FMRP/ USP. All animals were maintained at 25˚C. Food and water were provided ad libitum. The animals were anesthetized with ketamine (50mg/kg) and xylazine (10mg/kg) intraperitoneally, and infected intranasally with 20 µL of SARS-CoV-2 $2x10^4$ PFU (Brazil/SPBR-02/2020 strain, as previously described [11] diluted in PBS. From day 0 to day 5, mice were evaluated for weight, temperature, and clinical score. The clinical score was measured following the previously described [47]. At the indicated time point, mice were euthanized, and lungs were collected for following analyses. For mortality curves, mice survival was monitored daily for 10 days after infection.

## Statistical analysis

For puncta quantification, all histological sections were viewed on a 63x objective for digitalizing random images using the LSM 780 system in the Axio Observer microscope, covering an area of about ~1.7 mm$^2$ of lung parenchyma analyzed per case. Manual counting of puncta and cells was blinded and performed using the acquired images. Morphometric analyzes were performed as described [48]. The quantification of expression by immunohistochemistry was performed by calculating the percentage of marked area, the scanned images were opened in the ImageJ software, using the IHQ Toolbox plugin, which consists of a semi-automatic color selection tool that selects the pixels positive for immunohistochemical marking, differentiating them from background and H&E marking, after selecting the positive pixels, the images were transformed into 8-bits and the area occupied by these tones was calculated. The distribution of the gene expression and biochemical marker and puncta count data was evaluated using the Shapiro–Wilk test; the data were analyzed using non-parametric Kruskal-Wallis, Mann-

Whitney test and Spearman correlation. The data as violin plot graphs show median and quartiles. Statistical analyzes were performed using the GraphPad PRISM 5.0 program, with p<0.05 being considered statistically significant. Heatmaps were constructed using the heatmap.2 function in the R program (Project for Statistical Computing, version 3.4.1), the hierarchical clustering method used a correlation distance measure with ward.D2 and Canberra analysis.

## Supporting information

**S1 Fig. Histopathological patterns in lung autopsy of COVID-19 patients.** (**A**) The proportion of lung parenchyma area (loss of airspace) of COVID-19 patients and uninfected controls (benign area of the lungs from adenocarcinoma patients). (**B-C**) Representative images of H&E stain showing the lung parenchyma. Scale bars 200 μm. Each dot in the figure represents the value obtained from each individual. P-values are shown in the figures comparing the indicated groups, as determined by Mann–Whitney test. Data are represented as violin plots with median and quartiles.
(PDF)

**S2 Fig. Non-significant correlation or viral loads and inflammasome activation in lethal cases of COVID-19 patients.** Spearman correlation of pulmonary viral load and inflammasome activation in 47 fatal COVID-19 patients. (**A**) Correlation of viral N2 with NLRP3 puncta per parenchyma area; (**B**) Correlation of viral E with NLRP3 puncta per parenchyma area; (**C**) Correlation of viral N2 with ASC puncta per parenchyma area; (**D**) Correlation of viral E with ASC puncta per parenchyma area. r and p-value are indicated in the figure.
(PDF)

**S3 Fig. Gene expression in lungs of COVID-19 patients.** Correlation matrix of inflammasome and inflammatory gene expression in lung autopsy of 47 COVID-19 patients (**A**). Colors indicate correlation scores, categorized as positive strong correlation ($r \geq 0.70$; red); moderate positive correlation ($0.50 \geq r \leq 0.70$; orange); weak positive correlation ($0.30 \geq r \leq 0.50$; yellow); negative strong correlation ($r \geq -0.70$; dark blue); negative moderate correlation ($-0.50 \geq r \leq -0.70$; blue) or negative weak correlation ($-0.30 \geq r \leq -0.50$; light blue). Only correlations with p<0.05 are represented in the correlation matrix. (**B-U**) Expression of mRNA in the lung autopsies of COVID-19 patients and uninfected controls (benign area of the lungs from adenocarcinoma patients). Selected genes were *Il6* (**B**), *Il10* (**C**), *Il17* (**D**), *Ifna1* (**E**), *Ifnb1* (**F**), *Ifng* (**G**), *Il4* (**H**), *Il2* (**I**), *Tnfa* (**J**), *Il1a* (**K**), *Il1b* (**L**), *Il18* (**M**), *I1ra* (**N**), *Nlrp3* (**O**), *Nlrc4* (**P**), *Nlrp1* (**Q**), *Aim2* (**R**), *Pycard* (**S**), *Casp1* (**T**), *Casp4* (**U**).
(PDF)

**S4 Fig. CXR evolution of patients from Cluster 1 and Cluster 2.** Analysis of the first and last Chest x-radiography (CXR) of 47 COVID-19 patients belonging to Cluster 1 (n = 15) and Cluster 2 (n = 31). (**A**) patients with reduced opacities (green), stability (blue), and increased opacities (red) comparing the fists and first CXR. Impaired analyses are shown in gray. Representative images of first and last CXR from two patients from Cluster 1, indicating stability in moderate and mild pneumonia (**B**) and two patients from Cluster 2, indicating worsening conditions in cases of severe pneumonia (**C**).
(PDF)

**S1 Table. List of primer sequences used for real time-PCR.**
(PDF)

**S1 Data. Supporting Information file containing data from the experiments.**
(XLSX)

## Acknowledgments

We would like to thank Maira Nakamura, Patricia Edivânia Vendruscolo, Amanda Zuin and Josiane Ludke for their technical support.

## Author Contributions

**Conceptualization:** Keyla S. G. de Sá, Luana A. Amaral, Eurico Arruda, Alexandre T. Fabro, Dario S. Zamboni.

**Data curation:** Keyla S. G. de Sá, Luana A. Amaral, Dario S. Zamboni.

**Formal analysis:** Keyla S. G. de Sá, Tamara S. Rodrigues.

**Funding acquisition:** Dario S. Zamboni.

**Investigation:** Keyla S. G. de Sá, Luana A. Amaral, Tamara S. Rodrigues, Camila C. S. Caetano, Amanda Becerra, Sabrina S. Batah, Felipe T. Lopes, Isadora M. de Oliveira, Letícia S. Lopes, Leticia Almeida, Caroline M. Mota, Samuel Oliveira, Danilo T. Wada, Marcel Koenigkam-Santos, Ronaldo B. Martins, Roberta R. C. Rosales, Alexandre T. Fabro.

**Methodology:** Keyla S. G. de Sá, Alexandre T. Fabro, Dario S. Zamboni.

**Project administration:** Dario S. Zamboni.

**Resources:** Alexandre T. Fabro, Dario S. Zamboni.

**Supervision:** Dario S. Zamboni.

**Validation:** Keyla S. G. de Sá.

**Visualization:** Keyla S. G. de Sá, Dario S. Zamboni.

**Writing – original draft:** Keyla S. G. de Sá, Dario S. Zamboni.

**Writing – review & editing:** Luana A. Amaral, Tamara S. Rodrigues, Camila C. S. Caetano, Amanda Becerra, Sabrina S. Batah, Felipe T. Lopes, Isadora M. de Oliveira, Letícia S. Lopes, Leticia Almeida, Caroline M. Mota, Samuel Oliveira, Danilo T. Wada, Marcel Koenigkam-Santos, Ronaldo B. Martins, Roberta R. C. Rosales, Eurico Arruda, Alexandre T. Fabro.

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
