## [Decision Letter · Decision Letter 0]

1 Jun 2023

Dear Dr. Zamboni,

Thank you very much for submitting your manuscript "Inflammasome activation and pulmonary viral loads define two distinct clinical outcomes in COVID-19" for consideration at PLOS Pathogens. As with all papers reviewed by the journal, your manuscript was reviewed by members of the editorial board and by several independent reviewers. Although both reviewers found merit in the study, there were also a number of significant concerns raised.  In light of the reviews (below this email), we would like to invite the resubmission of a significantly-revised version that takes into account the reviewers' comments. 

In particular, the authors must address previous findings showing an absence of inflammosome activation in endothelial cells and provide more experimental data to support their contradictory findings.  More data should also be provided to support a pathological consequence and this should include experimental or experimental model data, not just correlative findings.

We cannot make any decision about publication until we have seen the revised manuscript and your response to the reviewers' comments. Your revised manuscript is also likely to be sent to reviewers for further evaluation.

Sincerely,

R. Keith Reeves

Guest Editor

PLOS Pathogens

Alexander Gorbalenya

Section Editor

PLOS Pathogens

Kasturi Haldar

Editor-in-Chief

PLOS Pathogens

orcid.org/0000-0001-5065-158X

Michael Malim

Editor-in-Chief

PLOS Pathogens

orcid.org/0000-0002-7699-2064

Reviewer's Responses to Questions

**Part I - Summary**

Reviewer #1: Keyla et al. undertake an important study on lung autopsies from 47 individuals who succumbed to COVID-19 to identify underlying pathophysiological mechanisms of COVID-19 lethality. The authors identified several biomarkers of inflammation, with a focus on the NLRP3 inflammasome, that were elevated in tissues from COVID-19 compared to controls (benign regions of lung tissue from patients who died from lung adenocarcinoma). By co-staining tissues for markers of cell type and NLRP3 inflammasome activation, the authors suggest that inflammasome activation primarily occurs in monocytes/macrophages and endothelial cells, consistent with prior reports from experimental infections that inflammasome activation can occur in these cell types. Moreover, hierarchical clustering revealed that tissues from COVID-19 patients fell into two distinct groups: one characterized by lower inflammatory signatures, disseminated intravascular coagulation but higher viral loads, and one characterized by higher incidence of fibrosis and inflammation but lower viral loads. Collectively, the data suggest that COVID-19 deaths can occur from distinct processes, providing insights into disease etiology and potentially patient management.

The authors should be commended for completing this important study. However, my enthusiasm is somewhat diminished by some limitations of the current work. The entire manuscript is focused on analyses on post-mortem samples. While this led to some interesting observations (e.g., the increased incidence of NLRP3 inflammasome activation in macrophages/monocytes and endothelial cells and the identification of patient clusters), no additional studies were performed to verify these findings in experimental models of SARS-CoV-2 infection. As such, the authors’ findings are largely observational and correlative. Moreover, several prior studies on autopsy samples from patients that died from COVID-19 have been published (e.g., PMID: 34648357, 33033248, 33158808). References to these and other reports are largely lacking, and a meta-analysis, or at minimum, a narrative comparison of these prior studies with the current manuscript, which would allow for the authors to establish whether or not their major conclusions are generalizable, were not conducted. Finally, there are a few occasions where conclusions should be softened or contextualized. I have also made a few recommendations for additional analyses of existing data that may further inform some of the authors’ conclusions.

Reviewer #2: In this study, the authors examined lung autopsies from COVID-19 patients and found that inflammasome activation correlates with COVID-19 severity. They also observed two mRNA expression patterns in lethal cases.

However, this manuscript suffers several serious defects.

Multiple studies have demonstrated the activation of inflammasomes in COVID-19 patients and its association with disease severity and clinical outcome. (including a study from the authors’ own group)

Nature volume 606, pages576–584 (2022) (https://doi.org/10.1038/s41586-022-04702-4)

Nature volume 584, pages463–469 (2020) (https://doi.org/10.1038/s41586-020-2588-y)

J Exp Med (2021) 218 (3) (https://doi.org/10.1084/jem.20201707)

Same phenomenon is described again in Figure 2 of this manuscript, which should be attached as supporting information.

Furthermore, the authors claimed that inflammasomes are active in endothelial cells, but a study published last year also analyzing lung autopsy samples from COVID-19 patients, shows that there was no inflammasomes activation in endothelial cells. (Nature volume 606, pages576–584 (2022)) Therefore, further evidence is required to prove the presence of active inflammasomes in endothelial cells. More importantly, instead of discussing, the authors should provide experimental results concerning the pathological relevance of inflammasome activation in endothelial cells.

The title and the conclusion of this manuscript are both misleading. Besides mRNA of inflammasome- related genes, the mRNA of many critical cytokines, such as IFNa, IFNb, IFNg, etc, is low expressed in the Cluster 1 as well. But the authors did not provide any evidence showing that inflammasome activation is the driving force behind the expression of these cytokines. Thus, inflammasome activation cannot be used to define the two clusters, let alone clinical outcomes.

**Part II – Major Issues: Key Experiments Required for Acceptance**

Reviewer #1: (No Response)

Reviewer #2: (No Response)

**Part III – Minor Issues: Editorial and Data Presentation Modifications**

Reviewer #1: Minor comments:

- As described above, prior studies that performed similar analyses on lung autopsy samples from COVID-19 patients should be cited and discussed. If possible, a meta-analysis of published data to determine if the conclusions from this study are generalizable would greatly enhance the significance. It is appreciated that some of the analyses (e.g., inflammasome-specific biomarkers) are probably not available for cross-comparison.

- Many clinical terms and biomarkers are incompletely described, which limits the accessibility of the manuscript to broad readership.

- To that point, upon trying to look up clinical definitions, it is not clear to me that observed levels meet established clinical criteria for some claims (e.g., platelet counts as an indication of DIC). Information should be provided to describe absolute levels in reference to established clinical definitions.

- If such patient data exists, it would be highly beneficial to provide biomarker data for patients prior to COVID-19 onset to assess ‘baseline’ levels of individual patients.

- Figure 1: adding labels to define cell populations would be helpful

- Figure 1: to the best of my knowledge, NLRP3 is not expressed in lung epithelia, including pneumocytes. Could NLRP3 staining in PDPN+ and SFTPC+ cells be background?

- Line 148 (and elsewhere): when referring to inflammasome activation, the authors should be specific when they intend to refer to the NLRP3 inflammasome. Not all inflammasomes utilize ASC, and not all cell types express NLRP3. Moreover, other inflammasomes have been implicated in the response to SARS-CoV-2.

- The antibody used to stain for IL-1beta detects both the pro- and active p17 form. Thus, that data is largely uninformative.

- Line 176: the correlation between NLRP3/ASC staining and time of disease (Fig 2C and 2D) is pretty weak. The conclusion should be measured or contextualized.

- The conclusion from Supplementary Fig 5 is unclear. Some correlations exist, but transcriptional correlations do not capture the state of inflammasome signaling.

- Fig 3B and 3C: it is unclear to me why the data is stratified by different x-axis variables

- Line 242: the word ‘dictate’ in the heading is inappropriate

- Given that cluster 2 is characterized by vascular dysfunction, it seems reasonable to reanalyze the data from Fig 1I-P across samples from cluster 1 and cluster 2 to determine if the incidence of inflammasome activation in endothelial cells is higher in cluster 1.

- Although the mechanism ventilation data is negative data, it should still be presented in the Results, and then discussed in the Discussion.

- Line 337: the authors should be careful to not imply that cells with markers of SARS-CoV-2 were infected by the virus – as the authors discuss elsewhere, it’s possible, particularly for phagocytic cells, that spike and other indicators are due to virus uptake or engulfment as opposed to infection.

- Line 352: should also cite PMID: 36129453

- A better description of the criteria for scoring CXR in the Methods would be helpful.

Reviewer #2: (No Response)

PLOS authors have the option to publish the peer review history of their article (what does this mean?). If published, this will include your full peer review and any attached files.

Reviewer #1: No

Reviewer #2: No
---

## [Decision Letter · Decision Letter 1]

8 Nov 2023

Dear Dr. Zamboni,

Thank you very much for submitting your manuscript "Pulmonary inflammation and viral replication define distinct clinical outcomes in fatal cases of COVID-19" for consideration at PLOS Pathogens. As with all papers reviewed by the journal, your manuscript was reviewed by members of the editorial board and by several independent reviewers. In light of the reviews (below this email), we would like to invite the resubmission of a significantly-revised version that takes into account the reviewers' comments.

Although the Reviewer's found added strength in the new mouse experiments, the limited numbers and some other experimental issues still prevent this study from full impact. If the authors consider revising, increasing the number of mice to reach more meaningful rather than just trending data would be important.  Mention of the new mouse experiments should also be included in the Abstract and Author Summary.  Additionally, based on Reviewer recommendations, conclusions should be softened on the nature of post-mortem study samples, since the data as they stand do not directly support causation.

We cannot make any decision about publication until we have seen the revised manuscript and your response to the reviewers' comments. Your revised manuscript is also likely to be sent to reviewers for further evaluation.

Sincerely,

R. Keith Reeves

Guest Editor

PLOS Pathogens

Alexander Gorbalenya

Section Editor

PLOS Pathogens

Kasturi Haldar

Editor-in-Chief

PLOS Pathogens

orcid.org/0000-0001-5065-158X

Michael Malim

Editor-in-Chief

PLOS Pathogens

orcid.org/0000-0002-7699-2064

Although the Reviewer's found added strength in the new mouse experiments, the limited numbers and some other experimental issues still prevent this study from full impact. If the authors consider revising, increasing the number of mice to reach more meaningful rather than just trending data would be important

Reviewer's Responses to Questions

**Part I - Summary**

Reviewer #1: The authors satisfactorily responded to and amended the issues I had with their description of methodology and they appropriately contextualized claims and softened conclusions. I also agree with the authors that removing aspects of Figure 1 does not lower the impact or novelty.

The authors have now also included an analysis comparing SARS-CoV-2 infection of hACE2+ WT and Nlrp3-/- mice. As has been reported elsewhere, the authors found that Nlrp3-/- mice exhibit a reduced inflammatory disease compared to WT mice, based on observed differences in parenchyma area, survival, and overall clinical score. Other indicators, such as inflammatory cytokines and weight loss *trend* in a manner consistent with Nlrp3 inflammasome involvement in disease, but in the present study are not overwhelmingly convincing. It appears the data may represent a single experiment, although it is not clear from the legend or methods.

The original finding of interest is that individuals that died due to COVID-19 cluster into two groups, suggesting distinct etiology. One of these clusters had hallmarks of inflammasome-mediated pathogenesis. However, since these are post-mortem analyses, the role of Nlrp3 in the development and progression of disease is only speculative. So while the new data does provide a more direct assessment of the role of Nlrp3 in SARS-CoV-2-associated inflammatory pathogenesis in mice, it feels like a missed opportunity to further substantiate the post-mortem findings. For example, one might predict that the Nlrp3-/- mice (that do not survive) may have profiles similar to cluster 1, but analyses to compare readouts that stratified human cluster 1 and 2 in the mice other than fibrosis by H&E (e.g., gene expression, viral load, etc) were not performed. To that end, I strongly encourage the authors to be very clear about what can and cannot be gleaned from the mouse study.

Reviewer #2: The revised manuscript is primarily focused on characterizing two distinct patterns observed in lethal cases of COVID-19. By utilizing hACE2 transgenic mice, the authors further provided experimental evidence showing the involvement of NLRP3 inflammasome in COVID-19 aggravation, which could potentially recapitulate the observation in the two patient clusters in mouse model of COVID-19. While this revised version is of interest, I believe this manuscript could be considerably strengthened by addressing the comments below.

In all mice studies, each group should include a minimum of five mice, which results in more compelling data, particularly when dealing with significant variations/error, as seen in Fig6 H/I.

To better reflect the varying activation levels of NLRP3 in the two clusters of patients, mice of Nrp3+/- heterozygous should be included in the experiments of Fig6 B/C/D/E/F. In addition, Fig6 G/H/I should add the Nlrp3+/+ group.

It would not be unexpected that a mouse model cannot fully mimic patient pathophysiology. Therefore, results of other cytokines, such as IFNa/b, IL-1, IL-18 etc., should also be included in Fig6 to better define the responses in mice with different Nlrp3 expression levels.

Minor points

The caption of fig6 is a bit confusing, especially in the description of mice’s genetic background.

Line 112, use “mechanical ventilation” instead of the abbreviation “MV” when it first appears in the text.

**Part II – Major Issues: Key Experiments Required for Acceptance**

Reviewer #1: (No Response)

Reviewer #2: (No Response)

**Part III – Minor Issues: Editorial and Data Presentation Modifications**

Reviewer #1: Minor comments:

- Cytokine analysis for mouse studies is missing in the methods section.

- The concluding sentence of the results (line 289-292) should include the term “…affects the disease progression [in mice]…” and “…observed in [some] lethal cases of COVID-19.” i.e., it’s certainly not all cases, given the authors conclusions regarding cluster 1 and cluster 2.

Reviewer #2: (No Response)

PLOS authors have the option to publish the peer review history of their article (what does this mean?). If published, this will include your full peer review and any attached files.

Reviewer #1: No

Reviewer #2: No
---

## [Decision Letter · Decision Letter 2]

24 Apr 2024

Dear Dr. Zamboni,

We are pleased to inform you that your manuscript 'Pulmonary inflammation and viral replication define distinct clinical outcomes in fatal cases of COVID-19' has been provisionally accepted for publication in PLOS Pathogens.

Best regards,

R. Keith Reeves

Guest Editor

PLOS Pathogens

Alexander Gorbalenya

Section Editor

PLOS Pathogens

Michael Malim

Editor-in-Chief

PLOS Pathogens

orcid.org/0000-0002-7699-2064

Reviewer Comments (if any, and for reference):

Reviewer's Responses to Questions

**Part I - Summary**

Reviewer #1: The authors satisfactorily responded to my comments. In particular, the mouse data is now more robust, and their assessment of the mouse data relative to the human data is appropriate.

Reviewer #2: The revised manuscript has been notably improved, particularly with the solidification of in vivo data concerning Nlrp3 WT and KO mouse infected with SARS-CoV-2. I now support the acceptance of this manuscript.

**Part II – Major Issues: Key Experiments Required for Acceptance**

Reviewer #1: (No Response)

Reviewer #2: (No Response)

**Part III – Minor Issues: Editorial and Data Presentation Modifications**

Reviewer #1: (No Response)

Reviewer #2: (No Response)

PLOS authors have the option to publish the peer review history of their article (what does this mean?). If published, this will include your full peer review and any attached files.

Reviewer #1: No

Reviewer #2: No

---

## [Editor Report · Acceptance letter]

28 May 2024

Dear Dr. Zamboni,

We are delighted to inform you that your manuscript, "Pulmonary inflammation and viral replication define distinct clinical outcomes in fatal cases of COVID-19," has been formally accepted for publication in PLOS Pathogens.

Best regards,

Michael Malim

Editor-in-Chief

PLOS Pathogens

orcid.org/0000-0002-7699-2064